# The Impact of Japan's Discharge of Nuclear-Contaminated Water on Aquaculture Production, Trade, and Food Security in China and Japan

**Xiao Liang [1], Shilong Yang [1,\*], Zhichao Lou [1] and Abdelrahman Ali [1,2]**

[1] School of Economics and Management, South China Agricultural University, Guangzhou 510642, China; liangxiao2016@stu.scau.edu.cn (X.L.); 20232161005@stu.scau.edu.cn (Z.L.); aaa31@fayoum.edu.eg (A.A.)

[2] Department of Agricultural Economics, Faculty of Agricultural, Fayoum University, Fayoum 63514, Egypt

\* Correspondence: 20211161015@stu.scau.edu.cn

**Abstract:** The aquaculture and fisheries sectors play critical roles in promoting a global nutritious and climate-friendly food system. The Japanese government started implementing the discharge of nuclear-contaminated water (NCW) into the Pacific Ocean in August 2023, which was followed by stopping the import of seafood from Japan to ensure the safety of imported food for Chinese citizens. The discharge of NCW into the ocean by Japan will directly harm the marine ecological environment and the global ecosystem due to the importance of China as the largest producer, processor, and exporter of aquatic products (APs). This paper employs the Global Trade Analysis Project (GTAP) model to simulate the future impacts of discharging the NCW under three different scenarios. The results showed that discharging NCW will lead to a global decline in AP production and also has negative repercussions on the macroeconomic landscape. Japan will face the most significant negative impact on its national macroeconomy, e.g., Japan's GDP, total imports, total exports, household income, and social welfare will decrease by 2.18%, 3.84%, 8.30%, 2.61%, and $130.07 billion; similarly, for China, the decrease will be 0.03%, 1.21%, 0.08%, and $728.15 billion, respectively. If China's AP consumption decreases by 10% and 20%, it will result in protein deficits of 1.536 million tons and 3.132 million tons, respectively. Japan's deficit will reach 138,000 tons and 276,000 tons. This necessitates supplementation via the consumption of other protein-rich foods, posing a significant threat to the nutritional security of food in both China and Japan.

**Keywords:** Fukushima contaminated water; fish trade; GTAP model; aquaculture economics; international trade

## 1. Introduction

Aquatic products constitute a significant portion of the human diet; they contribute to approximately one-sixth of the global required animal protein [1]. In addition, they secure essential micronutrients for human health, such as omega-3 fatty acids, amino acids, minerals, and vitamins [2]. With the continuous improvement in living standards, the demand for APs has grown significantly [3,4]. Since the blue revolution started in 1960, the global consumption of AP has nearly tripled, with the per capita consumption increasing from 9.0 kg in 1961 to 20.2 kg in 2020, exhibiting an annual growth rate of 3.2% [5].

Discharging nuclear-contaminated water (NCW) into the Pacific Ocean was officially announced by the Japanese government on 13 April 2021, with an expected timeline of 40 years for the full discharge of NCW [6–8]. Since this time, this topic has become a focal point of international concern to examine its socioeconomic and environmental impacts on the national as well as the global economy in the coming years [9,10]. In March 2011, many fishing ports were damaged as a result of a massive earthquake and tsunami that struck the Tohoku area of Japan; this earthquake and tsunami also caused severe damage to the Fukushima Daiichi Nuclear Power Plant (FNPP) and increased the contamination

to exceed the Japanese regulatory limit (it was 57.1%), but it decreased by 2015 to reach zero [11,12]. Japan officially began releasing accumulated radioactive water containing various radioactive elements into the ocean. The plan involves discharging 1.34 million tons of NCW into the ocean over a period of 30 years, and this action poses a direct threat to the marine and global ecosystems [2,6]. Additionally, radioactive elements, including Cesium-137, in the wastewater can accumulate in marine organisms, including fish, and then eventually be transmitted to the human body via the food chain [13,14].

Seafood provides 15% of the average animal protein intake for 2.9 billion people around the world. By 2050, it is projected that the contribution of marine food supply will be increased by 21–44 million tons (36–74% compared with the current production) to achieve sustainable food system transformation [15]. Nonetheless, the potential increase could be obscured by the risk of marine pollution (resulting in seafood contamination), such as the NCW, Mexico Oil Spill, etc., [16]. But the residual radioactive elements in the wastewater pose harm to marine life, potentially causing genetic mutations and resulting in a decline in fisheries' production quantity and quality. Moreover, consumers may be less confident in APs from contaminated regions, leading to a decrease in purchases, a shift toward products from uncontaminated areas, or even a cessation of AP consumption altogether [17–20]. These factors together will cause a shift in international trade and may also lead to a change in the price of APs across the regions in the future, which has not yet been addressed [21].

The General Administration of Customs of China has decided to temporarily halt the importation of aquatic products (including edible aquatic animals) from Japan, effective from 24 August 2023. This measure was taken in response to Japan's recent discharge of contaminated water from the Fukushima nuclear plant into the sea. By doing so, we aim to mitigate potential risks associated with radioactive contamination and safeguard the health and safety of Chinese consumers [22]. Simultaneously, China's Hong Kong and Macau have also announced an immediate ban on the import of APs from 10 prefectures (Tokyo, Fukushima, Chiba, Tochigi, Ibaraki, Gunma, Miyagi, Niigata, Nagano, and Saitama) in Japan, including all live, frozen, chilled, dried, or otherwise preserved APs, sea salt, as well as unprocessed or processed seaweed.

The global AP trade will be significantly affected by the discharging of the NCW of Japan via its direct and indirect impacts on the import–export quantity and value of relevant countries or regions [23]. Nevertheless, there is scarce literature that has examined these impacts on macroeconomic indicators, international trade, and food security. Qualitative analyses have shown that the discharge will affect global fisheries' production and processing, alter the international trade of relevant countries, and cause severe economic losses on a global scale [24,25]. Based on the current status of China's AP industry and its foreign trade competitiveness, discharging NCW will severely impact China's AP trade [26]. While few studies have estimated the effects of the NCW on the whole food systems, less attention has been paid to estimating NCW impacts on the international trade of APs and their impacts on food security. The quantity and quality of products produced from this contaminated water will be negatively affected and consequently reflected in the international trade and food security in the different regions, which have not yet been estimated. This is the target of the current study: to address the need to minimize the gap found in the literature. Mainland China is an important destination for Japan's AP exports [14]. According to the latest statistics of the Japanese Ministry of Agriculture, Forestry and Fisheries, from 2020 to 2022, the top four countries in terms of trading volume among the exporters of Japanese aquatic products are China, the United States, Vietnam, and South Korea. China's AP imports from Japan totaled $3.3 billion, accounting for 42 percent of Japan's total AP exports in 2022. Therefore, this paper aims to conduct a simulation analysis of the potential impact of Japan's discharge of NCW on the AP trade with China and its consequences for Japan's aquaculture and agriculture industries. The novelty of the current work can be highlighted in predicting the impacts of NCW on aquaculture production and international trade between China and Japan.

## 2. Methodology and GTAP Model Specification

*2.1. GTAP Model Framework and Scenario Setup*

2.1.1. GTAP Model Framework

The GTAP model was developed by Purdue University in 1993, and it is a type of CGE model widely utilized in economic and trade policy [25,27]. The standard GTAP model comprises six main entities: households, government, private sector, production sector, World Bank, and the rest of the world. When a country or region's savings enter the World Bank, the World Bank determines the allocation of investment funds. Household and government expenditures originate from domestic producers and the rest of the world. Domestic producers engage in production activities using primary inputs and intermediate goods, where intermediate goods come from both domestic producers and foreign imports. The produced goods are then divided into domestic sales and exports [27,28].

Under all assumed conditions, the GTAP model first postulates the utility function of the household sector as a Cobb–Douglas production function, expressed as follows:

$$U = C \prod\nolimits_{U_i} B_i \tag{1}$$

In Equation (1), where ($U$) represents the utility of the household sector, ($C$) is the scale parameter, ($U_i$) is the utility of an individual household, and ($B_i$) is the allocation parameter for an individual household. The consumption of the household sector is composed of three main activities: private expenditure, government expenditure, and savings. The consumption behavior of the private sector can be represented by a constant difference of elasticities (CDE) function [27], denoted as:

$$G(z, u) = \sum_{i=1}^{N} B_i u^{b_i e_i} Z^{b_i} \tag{2}$$

In Equation (2), where ($z$) represents the standardized price, ($u$) is the utility function of the household sector, ($b_i$) denotes the substitution elasticity, and ($e_i$) represents the expansion elasticity. The utility function for the government sector, under the condition of utility maximization, is the Leontief production function. The production function for the domestic production sector is a nested CES production function, expressed as:

$$Y = \alpha \left( \sum_{i=1}^{j} \delta_i x_i^{-\beta} \right)^{\frac{-1}{\beta}} \tag{3}$$

Equation (3) represents the total output ($Y$), where $\alpha$ is the efficiency elasticity, $\delta_i$ is the distribution parameter for individual firm inputs, and $x_i$ represents the output level of a single firm. Therefore, the household sector, private sector, government sector, and domestic production sector collectively form the regional household sector in the GTAP structural framework. These sectors engage in two main economic behaviors: consumption and savings. Consumption goods come from domestic producers and producers in the rest of the world. The income and savings of the household, private, and government sectors are deposited in the World Bank, and the World Bank controls the flow of funds globally (i.e., investment). From the perspective of production factors, the production sector primarily utilizes raw production factors (there are five types in the GTAP model: land, capital, skilled labor, unskilled labor, and natural resources) for commodity production, with some being sold domestically and some exported to the rest of the world.

The GTAP model conducts policy simulations based on given parameters, annual equilibrium prices, and quantities. The parameters of the CGE model are derived by solving for coefficients in all model equations using the coefficients and externally given elasticities from the GTAP database. This process of solving model parameters and coefficients is known as calibration. It is important to note that the GTAP model is calculated in value terms; hence, assuming constant quantities, the simulation and policy assessment

in the GTAP model rely heavily on prices. For instance, changes in tariff rates affect a country's consumption through price impacts, influencing its import quantity, social welfare, production structure, factor changes, and the trade of other countries. In the GTAP model, non-tariff barriers are also simulated by equivalent tariffs, and policy effects are reflected through prices. Subsidies to producers and consumers also affect prices through production taxes and indirect taxes. Therefore, prices are the core and key elements in conducting policy simulations using the GTAP model. Thus, this study will utilize the GTAP method to simulate and analyze the impact of Japan's discharge of nuclear wastewater on global seafood and other industries.

The GTAP 10 database comprises 141 countries and regions along with 65 industry sectors. Since the data in the GTAP 10 database were only updated until 2014, this study employs a method proposed by other researchers to extrapolate and update the database using exogenous variables such as GDP, capital stock, skilled labor, unskilled labor, and population growth rates. This extrapolation is conducted to align the database with the year 2022, ensuring that the model simulations closely reflect the current situation. The data required for dynamic extrapolation, including GDP, population, capital, and labor force, are sourced from the Global Trade Analysis Project (GTAP) at Purdue University.

For the purposes of model simulation and analysis, this study categorizes countries and industry sectors. The 141 countries are grouped into 14 regions, including China, China–Hong Kong, Japan, South Korea, the United States, and others. The 65 industry sectors are classified into agricultural and non-agricultural sectors, with further subcategories for the agricultural sector (19 categories as shown in Table A1).

2.1.2. Scenario Setting

The Japanese government started implementing the discharge of NCW from the FNPP into the ocean in August 2023 which was followed by stopping the import of seafood from Japan as a reaction taken by the different importer countries including China to ensure the safety of imported food for their local citizens. Thus, China, Hong Kong, and Macau have suspended the import of APs (including all live, frozen, chilled, dried, or otherwise preserved APs, sea salt, as well as unprocessed or processed seaweed, and edible aquatic animals) originating from Japan, starting from 24 August 2023 onwards.

The amount of APs imported from Japan to China has increased from $218 million in 2015 to $476 million in 2022. The proportion of this amount to Japan's total AP exports has also grown from 11.15% to 17.44%. According to reports from the Japan Broadcasting Corporation (NHK), since the leakage at the FDNPP, over 55 countries (regions) have imposed import restrictions on Japanese food. This article establishes a baseline scenario and four simulated scenarios considering the diffusion trajectory of NCW subsequent to its release by Japan and potential reactions from various nations.

Baseline Scenario: China Mainland, Hong Kong, and Macau prohibit the import of Japanese APs.

Simulated Scenario S1: Building upon the baseline scenario, AP production decreases by 10% in China Mainland, Japan, South Korea, the United States, Russia, and Canada.

Simulated Scenario S2: Building upon Simulated Scenario S1, technical trade barriers for APs increase by 10% in China Mainland, South Korea, and Russia.

Simulated Scenario S3: Building upon Simulated Scenario S2, AP production decreases by 10% in the United States, Latin American countries, Australia, New Zealand, and ASEAN countries. Technical trade barriers for APs increase by 10%.

Simulated Scenario S4: Building upon Simulated Scenario S3, the AP output of the United States, Latin American countries, Australia, New Zealand, and ASEAN countries is reduced by 10%, while technical trade barriers for APs increase by 10%.

In this paper, the UN Comtrade harmonized system (HS) is employed to classify APs, primarily including codes 0301 (live fish), 0302 (fresh or chilled fish), 0303 (frozen fish), 0304 (fish fillets and other fish meat), 0305 (smoked or salted fish), 0306 (crustaceans), 0307

(mollusks), 0308 (other shellfish and aquatic invertebrates), 1504 (fish fats and oils), 1604 (fish roe), and 1605 (prepared or preserved crustaceans and mollusks).

## 3. Results and Discussion

At present, APs have become one of the largest categories in terms of global production and trade, with approximately 96.57% of countries and regions engaging in activities related to the production and trade of APs. In the following section, the global AP production, consumption, and trade have been presented.

### 3.1. Global Aquatic Product Production

Global AP production is primarily concentrated in coastal countries along major oceans, including China, Indonesia, Vietnam, the Philippines, Japan, and South Korea along the Pacific coast; India and Bangladesh along the Indian Ocean coast; and the United States and Peru along the Atlantic coast. Over the years, as the contribution of APs to global food and nutritional security has grown, the production scale of global APs has steadily increased. From 2010 to 2020, global AP output increased from 165 million tons to 212 million tons, with an average annual growth rate of 2.57%. China, Indonesia, India, Vietnam, the United States, Russia, Peru, Bangladesh, the Philippines, and Japan are the top ten AP-producing countries globally. In 2020, the total production of APs in these ten countries reached 157 million tons, accounting for 73.88% of the world's AP output. China, as the world's largest AP producer, witnessed its production increase from 641.8 million tons in 2010 to 837.6 million tons in 2020, with an average annual growth of 2.71%, as shown in Figure 1 and Table A2. In 2020, China's production accounted for a significant 39.42% of the world's total output. The aquaculture industry also serves as a pivotal agricultural sector in Southeast and South Asian countries. In 2020, Indonesia, Vietnam, and the Philippines produced 23.39 million tons, 7.88 million tons, and 4.37 million tons of APs, respectively, contributing 16.78% to the world's total production. Indonesia experienced the fastest growth in AP production among Southeast Asian countries, with an average annual growth rate of 7.56% from 2010 to 2020. From a South Asian perspective, India and Bangladesh produced 13.27 million tons and 4.38 million tons of APs in 2020, ranking as the third and eighth largest AP producing countries globally.

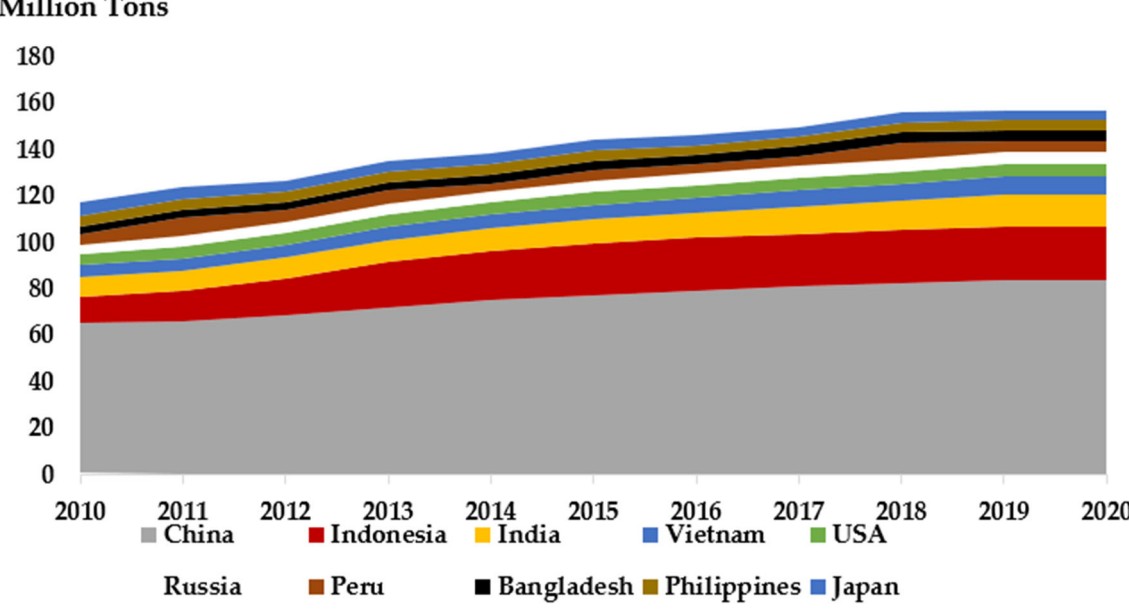

**Figure 1.** Aquatic product production in the top ten producing countries worldwide (2010–2020).

### 3.2. Mainly Fish in Global Aquatic Product Production

APs encompass aquatic animals and plants produced through marine and freshwater fisheries. Based on the International Standard Statistical Classification of Aquatic Animals and Plants (ISSCAAP) constructed by the FAO, aquatic products are categorized into seven main groups, namely freshwater fish, migratory fish, marine fish, crustaceans, mollusks, aquatic plants, and other aquatic organisms. In recent years, global freshwater aquaculture has experienced rapid development, maintaining its position as the largest category of AP production worldwide. From 2010 to 2020, the production of freshwater fish increased from 46.95 million tons to 65.49 million tons, with an average annual growth rate of 3.07%, consistently representing around 28% of the total global AP output. Following closely is marine fish, with relatively stable production over the years. The average production from 2010 to 2020 was 45.38 million tons, accounting for approximately 25.83% of the total global fish production. Additionally, the production of aquatic plants increased from 21.27 million tons in 2010 to 35.87 million tons in 2020, with an average annual growth rate of 5.46%, making it the fastest-growing category of APs, Table A3.

### 3.3. Growth in Global Aquatic Products Consumption

In recent years, due to the continuous increase in income levels and changing consumption patterns, there has been a consistent growth in the global demand for APs. From 2010 to 2020, the total global consumption of APs increased from 170 million tons to 217 million tons, with an average annual growth rate of 2.5%. Overall, the consumption preferences for APs have gradually formed over the course of history in major AP-producing nations due to their resource endowment advantages. As a result, the major consumers of APs align closely with the major producers. China, Indonesia, India, the United States, and Japan are the top five AP-consuming countries global. In 2020, these countries collectively consumed 141 million tons of APs, accounting for 64.84% of the total global AP consumption. China, being the leading consumer globally, witnessed an increase in consumption from 66.19 million tons in 2010 to 90.19 million tons in 2020, with an average annual growth rate of 3.16%. Its share of global AP consumption rose from 38.91% to 41.52% during this period, Figure 2 and Table A4.

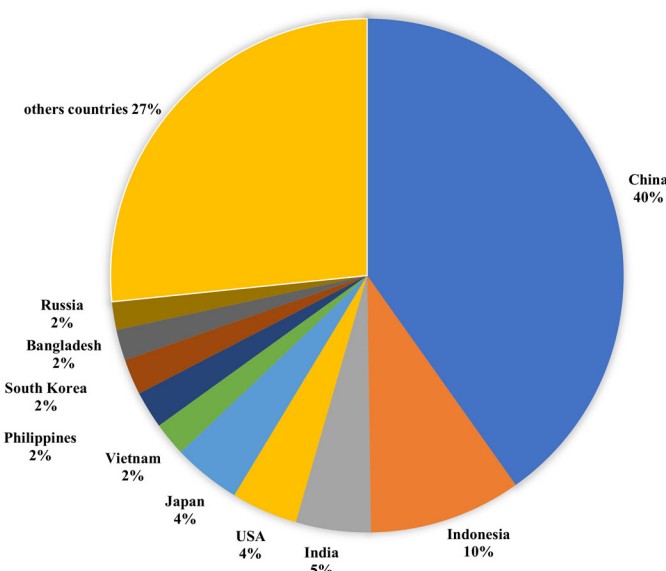

**Figure 2.** Share of the top ten aquatic products consumption worldwide as an average for the period (2010–2020).

### 3.4. Global Aquatic Product Trade

The global scale of AP trade has expanded as a result of the continuous improvement of aquaculture and fishing technologies worldwide, which has resulted in increasing

production and consumption levels of APs. From 2000 to 2021, the total volume, export volume, and import volume of global AP trade increased from 36.72 million tons, 16.58 million tons, and 20.14 million tons to 64.14 million tons, 32.63 million tons, and 31.51 million tons, with average annual growth rates of 1.86%, 1.98%, and 1.81%, respectively. In terms of AP exports, China, Norway, India, Chile, the United States, the Netherlands, Denmark, South Korea, Spain, and Morocco are the top ten AP-exporting countries in the world, Table 1. In 2021, the total export volume of these countries reached 14.67 million tons, accounting for 44.98% of the world's total exports. From 2000 to 2021, the Netherlands, the United States, Chile, and India showed the fastest growth in AP export volume, increasing from 0.45 million tons, 0.47 million tons, and 0.47 million tons to 1.18 million tons, 1.28 million tons, and 0.65 million tons, with average annual growth rates of 37.15%, 9.43%, and 5.52%, respectively.

Looking at AP imports, the main importing countries are mostly developed countries, with China, the United States, Japan, Spain, South Korea, France, Italy, Germany, the Netherlands, and Sweden being the top ten AP-importing countries in the world. In 2021, the total import volume of these countries reached 16.56 million tons, accounting for 52.56% of the world's total exports. From 2000 to 2021, the Netherlands and Sweden showed the fastest growth in AP import volume, increasing from 1.27 million tons, 1.62 million tons, and 0.64 million tons to 3.54 million tons, 3.09 million tons, and 0.90 million tons, with average annual growth rates of 7.56%, 3.75%, and 3.51%, respectively. Overall, China, the United States, the Netherlands, South Korea, and Spain are not only major exporting countries of APs but also major importing countries in the global AP trade. This can be attributed to their practice of importing raw materials while simultaneously re-exporting the processed products.

Aquatic products can be classified into low-value-added primary products and higher-value-added processed products based on their value addition. Products under Categories 0301–0305 are considered primary, while those under 0306–0308, 1504, 1604, and 1605 are classified as processed (Table A5). Currently, the global aquatic product trade is dominated by primary products, with the export volume of primary products accounting for 61.32% of the total AP exports in 2022. From 2000 to 2021, the export volume of primary products such as fresh and frozen fish, chilled fish, fish fillets, smoked and salted fish, and live fish increased from 10.28 million tons to 20.25 million tons, with an average annual growth rate of 3.44%. The import volume also grew from 13.25 million tons to 20.33 million tons, with an average annual growth rate of 2.16%. During the same period, the export volume of processed products such as crustaceans, mollusks, fish oil, fish roe, and fish paste increased from 7.10 million tons to 12.77 million tons, with an average annual growth rate of 2.98%. The import volume for processed products rose from 7.38 million tons to 12.13 million tons, with an average annual growth rate of 2.51%.

**Table 1.** Global aquatic product import and export situation (2000–2021) (unit: ten thousand tons).

| Export/Year | Total Ex. Volume | China | Norway | India | Chile | USA |
|---|---|---|---|---|---|---|
| 2000 | 1658.43 | 145.95 | 187.87 | 47.58 | 45.58 | 41.94 |
| 2005 | 2068.04 | 247.17 | 171.56 | 55.15 | 83.92 | 48.74 |
| 2010 | 2652.16 | 323.70 | 255.67 | 29.64 | 59.60 | 133.57 |
| 2015 | 2907.69 | 395.44 | 247.82 | 96.15 | 28.38 | 147.17 |
| 2016 | 2917.95 | 482.24 | 231.12 | 104.09 | 42.31 | 139.85 |
| 2017 | 3079.3 | 424.05 | 249.84 | 76.06 | 40.28 | 151.12 |
| 2018 | 3178.89 | 341.96 | 257.35 | 134.46 | 107.91 | 139.39 |
| 2019 | 3260.88 | 415.22 | 252.85 | 128.53 | 102.42 | 134.54 |
| 2020 | 3164.75 | 370.32 | 256.49 | 107.42 | 119.65 | 113.08 |
| 2021 | 3263.07 | 368.97 | 286.23 | 128.13 | 118.42 | 118.07 |
| Average | 2815.12 | 351.50 | 239.68 | 90.72 | 74.85 | 116.75 |

**Table 1.** *Cont.*

| Export/Year | Netherlands | Denmark | South Korea | Spain | Morocco | |
|---|---|---|---|---|---|---|
| 2000 | 67.31 | 70.82 | 47.27 | 79.78 | 31.06 | |
| 2005 | 91.77 | 76.44 | 33.67 | 46.22 | 32.42 | |
| 2010 | 79.59 | 73.02 | 67.24 | 100.24 | 41.88 | |
| 2015 | 96.77 | 78.06 | 52.06 | 102.87 | 53.76 | |
| 2016 | 103.55 | 81.19 | 48.72 | 103.94 | 55.9 | |
| 2017 | 111.33 | 82.13 | 40.43 | 110.85 | 58.56 | |
| 2018 | 115.61 | 79.84 | 47.32 | 111.92 | 61.46 | |
| 2019 | 114.21 | 81.01 | 52.05 | 107.26 | 62.40 | |
| 2020 | 108.69 | 84.45 | 46.81 | 104.33 | 68.46 | |
| 2021 | 113.1 | 89.11 | 65.33 | 111.42 | 68.89 | |
| Average | 100.19 | 79.61 | 50.09 | 97.88 | 53.48 | |
| **Import/Year** | **Total Im. Volume** | **China** | **USA** | **Japan** | **Spain** | **South Korea** |
| 2000 | 2014.21 | 127.62 | 162.54 | 309.42 | 129.05 | 68.55 |
| 2005 | 2318.62 | 196.43 | 191.78 | 282.63 | 153.06 | 118.04 |
| 2010 | 3337.87 | 257.41 | 236.43 | 227.30 | 104.56 | 116.00 |
| 2015 | 2842.75 | 276.13 | 251.78 | 215.42 | 159.86 | 131.22 |
| 2016 | 2895.57 | 200.14 | 257.80 | 210.94 | 163.99 | 135.88 |
| 2017 | 3049.85 | 300.02 | 265.12 | 218.30 | 169.03 | 127.30 |
| 2018 | 3176.19 | 344.65 | 274.91 | 208.12 | 165.02 | 146.24 |
| 2019 | 3145.52 | 449.80 | 268.36 | 210.55 | 168.96 | 142.67 |
| 2020 | 3326.63 | 391.82 | 275.65 | 192.64 | 156.77 | 142.99 |
| 2021 | 3151.56 | 354.48 | 309.10 | 195.68 | 149.63 | 154.24 |
| Average | 2925.88 | 289.85 | 249.35 | 227.10 | 151.99 | 128.31 |
| **Import/Year** | **French** | **Italy** | **Germany** | **Netherlands** | **Sweden** | |
| 2000 | 88.49 | 75.25 | 65.70 | 64.98 | 11.50 | |
| 2005 | 104.18 | 89.57 | 88.03 | 58.97 | 16.00 | |
| 2010 | 108.73 | 95.44 | 97.11 | 86.58 | 55.45 | |
| 2015 | 106.86 | 102.36 | 101.61 | 74.23 | 80.33 | |
| 2016 | 108.25 | 104.80 | 103.86 | 78.82 | 78.61 | |
| 2017 | 111.78 | 105.50 | 101.02 | 84.27 | 71.80 | |
| 2018 | 104.76 | 105.47 | 102.00 | 85.30 | 80.03 | |
| 2019 | 109.88 | 106.07 | 100.27 | 80.53 | 80.56 | |
| 2020 | 107.89 | 97.16 | 98.87 | 85.87 | 84.85 | |
| 2021 | 115.58 | 110.38 | 94.07 | 90.74 | 82.39 | |
| Average | 106.64 | 99.20 | 95.25 | 79.03 | 64.15 | |

Source of data: UN Comtrade. Note: The average values represent the average import and export volumes from 2000 to 2021.

## 4. Results of GTAP Model Simulation

*4.1. Macroeconomic Impact of the NCW Spread*

The global economy has a complex interrelationships, so the prohibition of Japanese AP imports, reduction in AP output, and the increase in technical trade barriers for APs will all have implications on GDP, total imports, total exports, household income, and social welfare in many countries besides Japan. Overall, the discharge of nuclear wastewater by Japan will result in adverse effects on the major economies worldwide through direct and indirect ways including the social welfare, global trade, food security, etc. From a social welfare perspective, China's prohibition of Japanese AP imports will lead to social welfare losses for Japan, China, and the EU countries, with Japan experiencing the greatest loss at $13.31 billion, followed by China with a welfare loss of $687.77 million. As the NCW spreads, the decline in global AP output will trigger major producing countries to tighten inspection and quarantine measures for AP imports. Similar to the baseline scenario, Japan faces the most significant negative impact on its national macroeconomy. Under Scenario S3, Japan's GDP, total imports, total exports, household income, and social welfare will

decrease by 2.18%, 3.84%, 8.30%, 2.61%, and $130.07 billion, respectively. Similarly, China's GDP, total imports, total exports, and social welfare will decrease by 0.03%, 1.21%, 0.08%, and $728.15 billion, respectively, as shown in Table 2.

**Table 2.** Macroeconomic changes in China, Japan, the United States, and the European Union under different scenarios.

| | GDP | Total Imports | Total Exports | Household Income | Social Welfare |
|---|---|---|---|---|---|
| China | | | | | |
| S1 | 0.02 | −1.16 | −0.05 | 0.05 | −6877.15 |
| S2 | 0.02 | −1.16 | −0.03 | 0.06 | −7216.11 |
| S3 | −0.03 | −1.21 | −0.08 | 0.00 | −7281.49 |
| Japan | | | | | |
| S1 | −2.14 | −3.78 | −8.26 | −2.57 | −13,312.75 |
| S2 | −2.14 | −3.78 | −8.26 | −2.57 | −13,337.58 |
| S3 | −2.18 | −3.84 | −8.30 | −2.61 | −13,006.75 |
| USA | | | | | |
| S1 | −0.13 | −0.17 | 0.11 | −0.13 | 1179.15 |
| S2 | −0.13 | −0.17 | 0.11 | −0.13 | 1210.32 |
| S3 | −0.18 | −0.24 | 0.12 | −0.18 | 1130.11 |
| EU | | | | | |
| S1 | −0.15 | −0.17 | −0.09 | −0.15 | −1171.39 |
| S2 | −0.14 | −0.17 | −0.09 | −0.14 | −1141.03 |
| S3 | −0.18 | −0.22 | −0.13 | −0.19 | −908.40 |

Note: Changes in GDP, total imports, total exports, and household income are in percentage (%); changes in social welfare are in million USD.

### 4.2. The Impact of NCW Spread on the Chinese and Japanese Agricultural Imports and Exports

From the perspective of China's major agricultural sectors, the prohibition of importing Japanese APs due to the discharge of NCW, coupled with a global reduction in AP production and increased technical trade barriers, and the results of the GTAP simulations revealed that China's AP imports in the three scenarios will significantly decrease (S1, S2, and S3) by 729.13%, 730.21%, and 731.20%, respectively. Additionally, due to the protein-rich content of APs and to ensure domestic food and nutritional security, China will increase imports in other protein-rich agricultural sectors. In the S1 scenario, imports of all meats, raw milk and dairy products, and other foods in China will increase by 0.21%, 0.23%, and 5.51%, respectively, and this consistent with the results of the recent literature. Meanwhile, the imports of other major agricultural sectors in China will significantly decline, such as imports of rice, wheat, and other cereals which will decrease by 2.54%, 2.15%, and 1.10%, respectively. Regarding China's major agricultural sectors' exports, as China primarily exports primary agricultural products, the increase in trade restrictions on APs globally will facilitate China's AP exports. In the S1, S2, and S3 scenarios, China's AP exports will increase by 28.20%, 28.93%, and 12.20%, respectively, as a substitution for the Japanese products. Meanwhile, exports of other major agricultural sectors in China will also increase. Exports of vegetable oils, oilseeds, and sugar crops will increase by 9.33%, 3.42%, and 2.72%, respectively, as shown in Table 3.

From the perspective of Japan's AP imports and exports, Japan's AP trade will suffer severe setbacks, with import and export declines expanding by several tens of times. In Scenarios S1, S2, and S3, Japan's AP imports will significantly decrease by 11.14 times, 11.13 times, and 11.20 times, respectively, and exports will decrease by 75.92 times, 75.92 times, and 76.06 times, respectively, consistent with the results of the recent literature. This could be a result of bans in other countries of the Japanese products or due to the decrease in the demand of the AP between many high consuming nations, as for example in China many people reported that they will either reduce the quantity or replace the AP with other animal protein sources to avoid the negative impacts of contamination on their heath.

Regarding Japan's imports in other industry sectors, due to the significant reduction in AP imports and the high demand for APs in Japanese dietary habits coupled with limited agricultural resources, Japan will need to increase the import of other agricultural products to ensure domestic food safety, which will put additional pressure on the trade balance to secure foreign currency for these imports. Imports of major agricultural products, such as rice, wheat, other cereals, oilseeds, sugar crops, and sugar, will increase substantially, with growth rates of 30.61%, 43.90%, 40.44%, 36.75%, 29.25%, and 22.11% in Scenario S1. Looking at exports from Japan's other industry sectors, the increased global anxiety about Japanese agricultural products due to nuclear radiation will lead to a significant decline in exports from Japan's main agricultural industry sectors. The decline will be most pronounced in the export of rice, wheat, oilseeds, and sugar crops, with reductions of 39.10%, 31.94%, 20.98%, and 18.43%, respectively, as shown in Table 4.

**Table 3.** Changes in imports and exports of China's major agricultural sectors under different scenarios.

| Products | Imports Change | | | Exports Change | | |
|---|---|---|---|---|---|---|
| | S1 | S2 | S3 | S1 | S2 | S3 |
| Rice | −2.54 | −2.65 | −3.41 | 1.76 | 1.89 | 4.57 |
| Wheat | −2.15 | −2.22 | −2.45 | 2.05 | 2.15 | 2.13 |
| Other Grains | −1.10 | −1.14 | −1.23 | 4.00 | 3.98 | 3.94 |
| Vegetables and Fruits | −1.23 | −1.28 | −1.52 | 1.35 | 1.38 | 1.95 |
| Oilseeds | −0.63 | −0.65 | −0.66 | 3.42 | 3.47 | 3.65 |
| Sugar Crops | −1.45 | −1.51 | −1.52 | 2.72 | 2.77 | 2.92 |
| Fiber Crops | −0.61 | −0.62 | −0.65 | 1.01 | 1.04 | 0.97 |
| Other Agricultural Products | −1.18 | −1.22 | −1.30 | 2.99 | 3.06 | 3.51 |
| All Meat | 0.21 | 0.23 | 0.23 | −6.44 | −6.48 | −6.89 |
| Wool | −0.58 | −0.59 | −0.61 | 0.64 | 0.69 | 0.59 |
| Aquatic Products | −729.13 | −730.21 | −731.20 | 28.20 | 28.93 | 12.20 |
| Vegetable Oils and Fats | −1.21 | −1.26 | −1.55 | 9.33 | 9.37 | 9.74 |
| Raw Milk and Dairy Products | 0.23 | 0.25 | 0.38 | −0.68 | −0.72 | −0.89 |
| Sugar | −1.62 | −1.68 | −1.82 | 1.58 | 1.63 | 2.51 |
| Other Foods | 5.51 | 5.70 | 6.03 | −20.24 | −20.48 | −22.22 |
| Tobacco and Beverages | −0.20 | −0.20 | −0.23 | 0.49 | 0.49 | 0.40 |
| Other Industries | −0.38 | −0.39 | −0.44 | 0.19 | 0.21 | 0.20 |

Note: The unit for import and export changes is percentage (%).

**Table 4.** Changes in imports and exports of Japan's major agricultural industry sectors in different scenarios.

| | Imports Change | | | Exports Change | | |
|---|---|---|---|---|---|---|
| | S1 | S2 | S3 | S1 | S2 | S3 |
| Rice | 30.61 | 30.59 | 30.50 | −39.10 | −39.08 | −38.3 |
| Wheat | 43.90 | 43.89 | 44.05 | −31.94 | −31.93 | −31.63 |
| Other Grains | 40.44 | 40.43 | 40.56 | −9.61 | −9.61 | −9.47 |
| Vegetables and Fruits | 12.11 | 12.11 | 12.09 | −9.61 | −9.62 | −9.40 |
| Oilseeds | 36.75 | 36.73 | 36.88 | −20.99 | −20.98 | −20.72 |
| Sugar Crops | 29.25 | 29.25 | 29.39 | −18.43 | −18.43 | −18.34 |
| Fiber Crops | 4.63 | 4.63 | 4.61 | −0.88 | −0.88 | −0.92 |
| Other Agricultural Products | 8.06 | 8.06 | 8.03 | −15.85 | −15.85 | −15.32 |
| All Meat | −20.69 | −20.68 | −20.86 | 86.30 | 86.26 | 86.61 |
| Wool | 7.10 | 7.10 | 7.05 | 34.04 | 33.99 | 34.3 |
| Aquatic Products | −1114.11 | −1113.72 | −1120.26 | −7592.06 | −7592.05 | −7605.66 |
| Vegetable Oils and Fats | 26.38 | 26.37 | 26.37 | 2.22 | 2.20 | 2.41 |
| Raw Milk and Dairy Products | −3.23 | −3.23 | −3.18 | 16.43 | 16.43 | 16.20 |
| Sugar | 22.11 | 22.1 | 22.02 | −4.15 | −4.16 | −3.64 |
| Other Foods | −72.89 | −72.87 | −72.87 | 157.5 | 157.51 | 156.43 |
| Tobacco and Beverages | −0.88 | −0.88 | −0.91 | 0.11 | 0.12 | 0.01 |
| Other Industries | 0.61 | 0.62 | 0.57 | −2.31 | −2.31 | −2.33 |

Note: The unit for import and export changes is percentage (%).

Currently, APs have become a significant source of food and nutritional security for residents in both China and Japan, and the discharge of NCW threats their food and nutrition security. China is the world's largest consumer of APs, with a seafood consumption exceeding 90 million tons in 2021, including 65 million tons of seafood, accounting for 45% of the global seafood consumption. The potential discharge of NCW by Japan is likely to alter the dietary habits of Chinese residents, significantly reducing the consumption of seafood and even APs, directly threatening China's food security. Simultaneously, it will impact Japan's AP market, leading to a significant increase in Japan's protein deficit. Calculations show that if China's AP consumption decreases by 10% and 20%, it will result in protein deficits of 1.536 million tons and 3.132 million tons, respectively. Japan's deficit will reach 138,000 tons and 276,000 tons, respectively. This necessitates supplementation through the consumption of other protein-rich foods, posing a significant threat to the nutritional security of food in both China and Japan, as shown in Table 5.

**Table 5.** Additional food consumption required after decrease in seafood consumption (unit: ten thousand tons).

| | The situation of additional food consumption needed to compensate for a 10% reduction in aquatic product consumption | | | | | | | |
|---|---|---|---|---|---|---|---|---|
| | Protein Gap | Eggs | Milk | Pork | Beef | Lamb | Chicken | Duck |
| China | 156.60 | 1195.42 | 4745.45 | 756.52 | 692.92 | 763.90 | 771.43 | 1010.32 |
| Japan | 13.80 | 105.33 | 418.13 | 66.66 | 61.05 | 67.31 | 67.97 | 89.02 |
| | The situation of additional food consumption needed to compensate for a 20% reduction in aquatic product consumption | | | | | | | |
| | Protein Gap | Eggs | Milk | Pork | Beef | Lamb | Chicken | Duck |
| China | 313.20 | 2390.84 | 9490.91 | 1513.04 | 1385.84 | 1527.80 | 1542.86 | 2020.65 |
| Japan | 27.60 | 210.66 | 836.25 | 133.32 | 122.11 | 134.62 | 135.94 | 178.04 |

## 5. Conclusions

Despite the importance of marine waters as a supplier for aquatic products (APs), having produced 112 million tons (63% of the global production) in 2020, they face a risk of nuclear contamination which threatens global food safety and security in the coming years. The discharge of NCW into the ocean by Japan will directly harm the marine ecological environment and the global ecosystem. Radioactive elements in the NCW will precipitate in marine organisms and spread through strong ocean currents to all marine regions, resulting in a reduction in marine AP yields. Ultimately, this will pose risks to human health through the food chain. The current results indicate that as the production decreases in major consumer and producer countries of APs, and technical trade barrier measures increase, the social welfare, GDP, and other macroeconomic indicators of both China and Japan will be negatively impacted. The import and export of Japanese APs will decrease at a rate several times faster, and concurrently, Japan's reliance on imports, particularly in the agricultural sector and especially for bulk agricultural products, will significantly increase. China's import of APs will be affected, leading to an increase in the import of other meat protein sources. Thus, we conclude that discharging the NCW will cause negative impacts on China and Japan's macroeconomic and AP trade not only on both countries but also with these impacts projected to move across the borders.

**Author Contributions:** X.L. conceived, designed, and conducted this study. S.Y. revised the manuscript. Z.L. and A.A. were involved in the analysis and interpretation of the data and funded this study. All authors have read and agreed to the published version of the manuscript.

**Funding:** This research was funded by The National Natural Science Foundation of China Project: The Economic Transformation and the Development of Regional Agricultural Products Value Chain of ASEAN and China (No. 71961147002).

**Institutional Review Board Statement:** Not applicable.

**Informed Consent Statement:** Not applicable.

**Data Availability Statement:** All the data are obtained from the General Administration of Customs of the People's Republic of China, GTAP database, UN Comtrade. They are available on request from the corresponding author.

**Acknowledgments:** The authors acknowledge the support provided by their respective institutions.

**Conflicts of Interest:** The authors declare no conflicts of interest.

## Appendix A

**Table A1.** GTAP model classification of countries and sectors.

| Countries/Regions | Sectors | Categories |
|---|---|---|
| China; Hong Kong, China; Japan; South Korea; USA; Russia; Canada; Australia; New Zealand; ASEAN countries; European Union (27 Countries); UK; Latin American countries; Other countries in the world | Agricultural sectors | Paddy rice; Wheat; Other cereals (Indica rice); Vegetables, fruits, and nuts; Oil crops; Sugar crops; Fiber crops; Other crops; Cattle, sheep, horses; Other animal products; Wool; Aquatic products; Beef products; Other meat products; Vegetable oils and fats; Raw milk and dairy products; Sugar; Other food products; Tobacco and beverages |
| | Non-agricultural sectors | Other industries |

**Table A2.** Aquatic product production in the top ten producing countries worldwide (2010–2020) (unit: million tons).

| Year | World Total Production | China | Indonesia | India | Vietnam | USA | Russia | Peru | Bangladesh | Philippines | Japan |
|---|---|---|---|---|---|---|---|---|---|---|---|
| 2010 | 165.05 | 64.18 | 11.61 | 8.51 | 4.95 | 4.81 | 4.20 | 4.41 | 3.04 | 5.05 | 5.34 |
| 2011 | 173.12 | 65.81 | 13.65 | 8.01 | 5.22 | 5.58 | 4.39 | 8.37 | 3.12 | 4.83 | 4.79 |
| 2012 | 176.61 | 68.94 | 15.45 | 9.11 | 5.59 | 5.43 | 4.48 | 4.95 | 3.26 | 4.75 | 4.84 |
| 2013 | 184.60 | 72.11 | 19.41 | 9.22 | 5.80 | 5.53 | 4.52 | 6.01 | 3.41 | 4.57 | 4.76 |
| 2014 | 189.62 | 75.22 | 20.86 | 9.89 | 6.05 | 5.41 | 4.43 | 3.72 | 3.55 | 4.58 | 4.75 |
| 2015 | 195.05 | 77.47 | 22.35 | 10.13 | 6.38 | 5.47 | 4.62 | 4.94 | 3.68 | 4.50 | 4.59 |
| 2016 | 197.23 | 79.49 | 22.53 | 10.78 | 6.71 | 5.35 | 4.95 | 3.93 | 3.88 | 4.23 | 4.34 |
| 2017 | 204.02 | 81.10 | 22.63 | 11.63 | 7.11 | 5.48 | 5.07 | 4.29 | 4.13 | 4.13 | 4.30 |
| 2018 | 210.87 | 82.19 | 23.01 | 12.52 | 7.51 | 5.28 | 5.33 | 7.31 | 4.28 | 4.35 | 4.30 |
| 2019 | 212.46 | 83.76 | 23.39 | 13.27 | 7.88 | 5.35 | 5.24 | 5.01 | 4.38 | 4.37 | 4.30 |
| 2020 | 212.46 | 83.76 | 23.39 | 13.27 | 7.88 | 5.35 | 5.24 | 5.01 | 4.38 | 4.37 | 4.30 |
| Average | 192.83 | 75.82 | 19.84 | 10.58 | 6.46 | 5.37 | 4.77 | 5.27 | 3.74 | 4.52 | 4.60 |

Source of data: FAOSTAT. Note: The data for the year 2020 are based on FAO projections.

**Table A3.** Aquatic product production by category worldwide (2010–2020) (unit: million tons).

| Year | Freshwater Fish | Migratory Fish | Marine Fish | Crustaceans | Mollusks | Aquatic Plants | Other Aquatic Organisms |
|---|---|---|---|---|---|---|---|
| 2010 | 46.95 | 20.34 | 43.60 | 11.35 | 20.29 | 21.27 | 1.25 |
| 2011 | 48.38 | 20.39 | 47.78 | 11.76 | 20.60 | 22.92 | 1.30 |
| 2012 | 51.36 | 20.52 | 44.28 | 12.12 | 21.18 | 25.83 | 1.31 |
| 2013 | 54.26 | 21.22 | 44.10 | 12.50 | 21.76 | 29.30 | 1.44 |
| 2014 | 56.27 | 21.66 | 43.27 | 13.31 | 23.38 | 30.22 | 1.52 |
| 2015 | 58.11 | 21.85 | 44.49 | 13.68 | 23.26 | 32.14 | 1.53 |
| 2016 | 60.20 | 22.02 | 43.88 | 14.09 | 22.79 | 32.74 | 1.50 |
| 2017 | 62.66 | 23.28 | 44.90 | 15.19 | 23.68 | 32.92 | 1.38 |
| 2018 | 63.86 | 22.41 | 49.45 | 15.72 | 23.58 | 34.37 | 1.49 |
| 2019 | 65.49 | 22.77 | 46.72 | 16.15 | 23.90 | 35.87 | 1.57 |
| 2020 | 65.49 | 22.77 | 46.72 | 16.15 | 23.90 | 35.87 | 1.57 |
| Average | 57.55 | 21.75 | 45.38 | 13.82 | 22.57 | 30.31 | 1.44 |

Source of data: FAOSTAT. Note: The data for the year 2020 are based on FAO projections.

**Table A4.** Aquatic product consumption in the top ten consuming countries worldwide (2010–2020) (unit: million tons).

| Year | Total Consumption | China | Indonesia | India | USA | Japan | Vietnam | Philippines | South Korea | Bangladesh | Russia |
|------|------|------|------|------|------|------|------|------|------|------|------|
| 2010 | 170.10 | 66.19 | 10.81 | 7.61 | 7.81 | 9.59 | 3.52 | 5.01 | 3.73 | 3.01 | 3.70 |
| 2011 | 180.00 | 67.70 | 12.98 | 7.02 | 8.08 | 8.97 | 3.44 | 4.76 | 4.23 | 3.07 | 3.71 |
| 2012 | 178.35 | 72.11 | 14.52 | 8.12 | 7.87 | 9.16 | 3.65 | 4.73 | 4.09 | 3.24 | 3.97 |
| 2013 | 188.08 | 73.72 | 18.51 | 8.25 | 8.01 | 8.43 | 3.42 | 4.46 | 3.96 | 3.41 | 3.83 |
| 2014 | 192.74 | 77.54 | 19.82 | 8.67 | 7.88 | 8.63 | 3.42 | 4.61 | 4.38 | 3.62 | 3.54 |
| 2015 | 200.43 | 80.28 | 21.44 | 9.02 | 8.22 | 8.29 | 4.02 | 4.70 | 4.46 | 3.91 | 3.06 |
| 2016 | 201.56 | 81.37 | 21.64 | 9.72 | 8.33 | 7.79 | 4.14 | 4.40 | 4.49 | 4.06 | 3.14 |
| 2017 | 206.91 | 85.21 | 21.85 | 9.96 | 8.53 | 7.93 | 4.66 | 4.34 | 5.03 | 4.33 | 3.24 |
| 2018 | 217.47 | 86.99 | 22.20 | 10.85 | 8.73 | 7.93 | 5.08 | 4.59 | 5.04 | 4.48 | 3.36 |
| 2019 | 217.23 | 90.19 | 22.47 | 11.70 | 8.56 | 7.93 | 5.44 | 4.66 | 4.64 | 4.61 | 3.41 |
| 2020 | 217.23 | 90.19 | 22.47 | 11.70 | 8.56 | 7.93 | 5.44 | 4.66 | 4.64 | 4.61 | 3.41 |
| Average | 197.28 | 79.23 | 18.97 | 9.33 | 8.23 | 8.41 | 4.20 | 4.63 | 4.43 | 3.85 | 3.49 |

Source of data: FAOSTAT. Note: Data for the year 2020 are based on FAO projections.

**Table A5.** World trade of primary and processed aquatic products (2010–2020) (unit: million tons).

| Year | Export Volume | | Import Volume | |
|------|------|------|------|------|
| | Raw Aquatic Products | Processed Aquatic Products | Raw Aquatic Products | Processed Aquatic Products |
| 2000 | 1028.91 | 710.61 | 1325.10 | 738.38 |
| 2005 | 1330.24 | 838.14 | 1518.89 | 858.97 |
| 2010 | 1708.01 | 964.05 | 2450.57 | 916.13 |
| 2015 | 1846.50 | 1061.34 | 1854.46 | 1006.69 |
| 2016 | 1844.20 | 1080.49 | 1964.35 | 992.27 |
| 2017 | 2047.00 | 1034.05 | 2074.24 | 1001.91 |
| 2018 | 2147.11 | 1033.23 | 2132.40 | 1074.89 |
| 2019 | 2107.22 | 1197.07 | 2030.07 | 1150.63 |
| 2020 | 2014.77 | 1197.48 | 2013.03 | 1338.84 |
| 2021 | 2025.03 | 1277.30 | 2033.15 | 1213.38 |

Source of data: UN Comtrade.

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
