# Peer review of "The Impact of Japan’s Discharge of Nuclear-Contaminated Water on Aquaculture Production, Trade, and Food Security in China and Japan"

_sustainability, doi:10.3390/su16031285_

Round 1

Reviewer 1 Report

Comments and Suggestions for Authors

1 Please add the full name of AP and GTAP in abstract

2 l39 "effect" should be "affect"

3 The English need to be improved and edited by a native English speaker. Some sentense's structure and tense is confused. For example, the sentense in l81-l84 only contains a subject and several appisitive, the sturcture of the sentense is incomplete. l96 "shown" should be "have shown"

4 the sturcture of introduction is really chaotic. The authors used a hole paragraph to introduce the current situation of globle aquaculture, but it is not relevant to the topic. However, the current situation of Chinese and Japanese fishery is defective. Besides, authors used redundant to describe the background of the policy and nuclear waste. In my opinion, these description need only be concluded by several sentense but not a detailed story. The highlight need to be rebuilt. In introduction, it could be named as purpose instead. The last paragraph is needless and should be deleted. So the introduction need to be modified and reconstucted to be scientific and logistic.

5 l125-134 Why locate this paragraph in this section? The introduction and comparation of the model should be placed in the introduction. And the appearence of the GTAP model is really abrupt. 

6 l211 (Li, 2023) is an uncorrect citation in this journal. And I think this section need to be moved to introduce

7 l237-245 the content of this paragraph is introduction like.

8 Section 3.1 makes me confused.  Is there any connection between the content and the title or topic? I am not skilled in your model, but my experience prompts me that this section need to be put in the introduction or deleted. Besides, there is no 3.2, why not use 3. in the title?

9 The conclusion is so looooong. The conclusion need to be shortened to no more than 1 paragraph.

Comments on the Quality of English Language

The manuscript needs to be edited by a native English speaker

Reviewer 2 Report

Comments and Suggestions for Authors

Authors captures current issues very well.

Abstract

1. Line 6-8, who subsequently stopped importing seafood from Japan to ensure the safety of imported food for local residents. stopping the import of seafood from Japan to ensure the safety of imported food for their local citizens?

2. Line 11, AP? The first time an abbreviation appears, it is recommended to list the full name.

3. Line 19-21, for China, it is indeed necessary to supplement by eating other protein-rich foods. But why does it pose a major threat to Japan's food nutrition security? Japan does not export to China, it should retain a lot. Why does it pose a major threat to Japan’s food nutrition security?

4. Line 21-23, the conclusion of this sentence is unclear. We recommend coming up with clear and feasible coping strategies.

Introduction

1. Typos on effect and tones.

2. Line 78-81, we don't understand the meaning of this sentence.

3. Line 81-87, we don't understand the meaning of this paragraph.

4. Line 116-123, this section is not needed. Recommended to delete.

Methodology and GTAP Model Specification

1. Line 125-134, some of the description in this paragraph is unnecessary. It is recommended to keep only the content related to Methodology and GTAP Model Specification, and delete unnecessary descriptions or rewrites.

2. Line 225-227 and Line 228-230, these two Simulated Scenario S3 seem to be the same. clarify.

Results and Discussion

1. Line 299, “globall” typo.

2. For Table 1, the countries in the columns should not be broken. It is recommended to change the format of Table 1.

3.We believe that discussion is insufficient.

Results of GTAP Model Simulation

1. The label of Table 2 that we did not see.

2. We believe there is a lack of policy analysis and discussion.

Conclusion

It is recommended to simplify.

Reviewer 3 Report

Comments and Suggestions for Authors

The article titled "Impact of Japan's Discharge Nuclear Contaminated Water on Aquaculture Production Trade and Food Security in China and Japan" is very interesting and well-written as below:

1) Originality: The article examines the socio-economic and environmental impacts of Japan's decision to discharge nuclear contaminated water (NCW) into the Pacific Ocean. It uses the GTAP model to simulate future impacts under various scenarios, highlighting potential global declines in aquaculture production and negative macroeconomic consequences; thus the article has high originality and academic interest.

2) Contribution to scholarship: The study particularly focuses on the significant impacts on Japan's and China's economies, including decreases in GDP, imports, exports, household income, and social welfare. Additionally, it discusses the potential protein deficits and the need for alternative protein-rich foods, underlining threats to nutritional security in both countries. Overall, the discussion and theoretical framework are sound and sensible.

3) Sound methodology: The paper's methodology involves a comprehensive simulation analysis using the GTAP model and offers insights into the broader implications of the NCW discharge on global trade, aquaculture economics, and food security.

4) Overall well-structured and well-written.

I would recommend publication.

Comments on the Quality of English Language

The quality of English language in the article appears to be adequate for academic standards. There are minor issues with sentence structure and phrasing, which could be attributed to non-native English writing. However, these issues do not significantly hinder the overall readability or comprehension of the text. The article maintains a formal and academic tone, suitable for its subject and intended audience. It follows standard academic conventions in terms of structure and presentation of information. Overall, the language quality is satisfactory for an academic publication, with room for minor improvements in language fluency and sentence construction.

Round 2

Reviewer 1 Report

Comments and Suggestions for Authors

No problem